# Comment on "A universally applicable method of calculating confidence bands for ice nucleation spectra derived from droplet freezing experiments" by Fahy, Shalizi and Sullivan (2022).

**Gabor Vali**

University of Wyoming, Laramie, WY. USA.

**Correspondence:** Gabor Vali (vali@uwyo.edu)

**Abstract.** The analysis methods in Fahy et al. (2022) and their interpretation of experiments with water drops containing ice nucleating particles raise some technical issues and prompt a discussion of the principles involved in the use of differential spectra.

## 1 Introduction

Fahy et al. (2022, F22) delves into the how best to derive ice nucleation spectra (spectra for short in the following) from drop freezing experiments. Among other issues, alternative data processing methods are discussed and a new method is presented for the calculation of confidence intervals. As the author of the paper that first introduced these spectra (Vali, 1971), I appreciate this development of the methods of analyses of the spectra. The results derived in F22 will undoubtedly prompt further advances in the understanding of freezing nucleation.

The purpose of this Comment is to show the difference in perspectives between that taken in F22 and that forming the basis of Vali (2071, 2019) for representing the results of freezing nucleation experiments. The impact of the data processing recommended in F22 is examined.

Helpful clarifications of the reasoning employed in F22 are given in Fahy and Sullivan (2023) and are incorporated into the discussion that follows. Even more detailed examination of minor points are in Vali (2023).

## 2 Two perspectives

The point of departure in F22 consists of three elements: (1) $k(T)$ should depict the underlying function representative of the activity of the INPs studied, (2) given experimental results approximate that function, and (3) $k(T)$ is continuous across the temperature range of the measurements. The first element arises from the desire to characterize INPs in a way that permits rigorous comparisons between experiments with different substances and different conditions. The second point is a direct consequence of limited sample sizes in any experiment, although that limitation is rapidly decreasing with progress in instrumentation and observational techniques. The main justifications for (3) is that experiments only sample from a probability distribution of potential nucleation temperatures for each INP and that nucleating sites can be active over a range of temperatures. To facilitate the discussion, this probability density function is designated as $P_{site}(T)$.

The first point listed above is the perspective that differentiate the work in F22 from the perspective represented by analyses in Vali (1971, 2019) and many other earlier publications where focus is on making $k(T)$ the representation of the observed freezing temperatures in as concise a form as possible. What distinguishes these two perspectives is when and how analysis/interpretation of observations enters. With $k(T)$ viewed as representation of empirical data, interpretations follow data analysis with considerations of experimental uncertainties and other relevant knowledge. With the spectra viewed as depictions of the underlying function describing the activity of a sample, the data analysis combines measurement results with independent knowledge (assumption) of the random effects that affect sites and which are incorporated in $P_{site}$.

The section to follow discusses the issue of data representation with fixed or variable bins widths in temperature. Then, the $P_{site}$ is discussed in Section 4and the question of continuity in $k(T)$ is examined in Section 5.

## 3   Differential spectra derivation

Basically, the spectra represent the results of counting freezing events that occur at different temperatures as a sample is cooled gradually from above $0^{\circ}$C until all sample drops are frozen or the cooing is stopped. For data representation purposes, the spectra equations can be viewed as summaries of the observations. Freezing temperatures of the drops are distinct events and the differential spectra represent that discreteness as best as the data and sample size allow. Freezing events are precise temperature values (apart from instrumental errors). The temperature at which a given site initiates freezing is taken to be the characteristic temperature $T_c$ of the active site. Further considerations (Section 4) extend this definition to a single realization from a distribution of temperatures about the characteristic temperature, but with a single experiment, the observed temperature is the best estimate available for $T_c$.

The differential spectrum is defined in Vali (1971) as

$$k(T) = -\frac{1}{X * dT} * ln(1 - \frac{dN}{N(T)}) \tag{1}$$

where $N(T)$ is the number of drops not frozen[1] at $T$ and $dN$ is the number freezing within the temperature interval $dT$ as the sample is cooled past $T$. The dimension of $k(T)$ is [cm$^{-3}$ $^{\circ}$C$^{-1}$] for $X = V$. The use of differentials for $dN$ and $dT$ underscore the intention that $k(T)$ reflect nucleation activity observed at $T$. This is an ideal that has to be abandoned for any finite sample size (total number of drops), so for practical use one has

$$k(T) = -\frac{1}{X * \Delta T} * ln(1 - \frac{\Delta N}{N(T)}) \tag{2}$$

with the interval within which the activity is observed expanded to $\Delta N$ and $\Delta T$. The point is that the purpose of the differential spectrum is to focus on activity at specific temperatures. The choice of the magnitude of $\Delta T$ is driven by a consideration of the interplay between wanting to avoid too many intervals with no freezing events, the greater uncertainty that results from smaller $\Delta N$ and the desire for higher temperature resolution. In most literature the range of $\Delta T$ values is 0.2 to $1.0^{\circ}$C and it is kept constant thorough the range of freezing temperatures observed in an experiment.

More discussion about about the choice of temperature interval is given in Section 4 of Vali (2019). In F22, to facilitate the application of a continuous function for $k(T)$, variable bin widths are used. The interval width $\Delta T$, for adjacent freezing events $T_i$, $T_j$, and $T_k$, is determined as

$$\Delta T = \frac{T_i - T_j}{2} - \frac{T_j - T_k}{2} \tag{3}$$

---

[1]F22 has an error in Section 2, defining N as the number already frozen

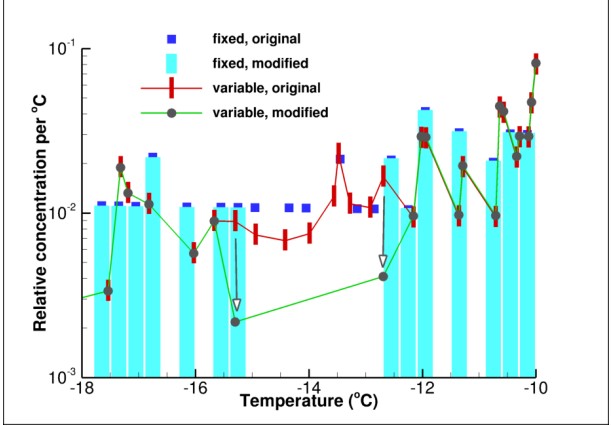

**Figure 1.** A segment of the differential spectrum $k(T)$ processed in four different ways. See the text for details.

for cases when one freezing event is observed at each temperature $T_i$ $T_j$ and $T_k$. If more that one event is associated with these temperatures then weighing factors are assigned according to the number of events for the temperature. This latter case arises from limitations in the resolution of the temperature measuring instrument of in the data recording system. Such a limitations constitute inherent binning of the data.

The use of the variable bin width resulting from Equation 3 has two consequences. It can produce point-to-point jumps in $k(T)$ (noise in a sense; Petters (2023)) which are subsequently smoothed. More importantly, this method creates a value for $k(T_j)$ that is dependent on its neighboring events $T_i$ and $T_k$. This is undesirable if the intention is to have $k(T)$ represent observed activity directly.

The effect arising with the use of variable bin widths can be elaborated with the help of an example. A somewhat extreme case is chosen. Fig. 1 shows a segment of the differential spectrum $k(T)$ which is shown in its totality in Fig. 4 in Vali (2019). Blue squares indicate the spectrum with $\Delta T = 0.3^{\circ}$C. The heavy vertical bars in red show the same data with intervals chosen as in Eq. 3. For purposes of illustration, 6 events of the original data between $-12.92^{o}$C and $-14.94^{o}$C were removed and the spectra recalculated. The bar diagram shows the new values with $\Delta T = 0.3$°C and the dark gray circles with Eq. 3. While the bar diagram and the blue squares remain in agreement, the two dark gray points either side of the gap in freezing events show a large decrease. These are indicated by vertical arrows. The magnitude of the decrease is near a factor of 4 in both cases. The same lowering of data points near gaps in the spectrum with variable $\Delta T$ can be seen, albeit to lesser degrees, at temperatures near $-16^{o}$C, $-11.4^{o}$C and $-10.7^{o}$C.

The alteration of $k(T)$ due to changes in neighboring freezing events is an undesirable for concise data representation. Even though the effect is minor for data with freezing

events closely spaced, there is a reasonable objection to the use of variable $\Delta T$ on the basis of principle. The fixed $\Delta T$ approach treats all data points with equally across the range of observations.

In all, the recommendation made in Vali (2019) for the use of fixed $\Delta T$ is repeated here, if the concise data representation is desired.

## 4    The $P_{site}$ function

In the foregoing section, observed temperatures are taken as
best estimates of the $T_c$ but it was also pointed out that random effects always enter into making any observed freezing event vary about what $T_c$ value would result from looking for the mode of a large number of repetitions. Those repetitions would lead to a distribution of freezing event, desig-
nated as $P_{site}$. For historical reasons it may be worth noting that this distribution was defined by Vali and Stansbury (1966) as a nucleation rate $P_1(T, T_c)$ where $T_c$ is the characteristic temperature associated with the site. The $P_{site}$ distribution would be the observed frequency of freezing events
resulting from the nucleation rate $P_1(T, T_c)$ per unit time.

Briefly, the fundamental reason for a degree of random variation in nucleation temperature on a site is the chaotic fluctuation of water molecules as they are forming and exiting ice embryos. Theoretical estimates for the resulting $P_{site}$
(for heterogeneous nucleation) are not reliable because of unknown properties of sites. A direct attempt to obtain a quantitative estimate of $P_{site}$ is given in Vali (2008, V08) along with the limitations of validity of that estimate. In V08, $P_{site}$ is assumed to be a Gaussian function with standard devia-
tions of 0.2 and $0.42^o$C for two different samples. From this, it was concluded that as a rough estimate observed freezing temperatures approximate $T_c$ within about $1^o$C. Other sources estimate this range to be larger.

For the analyses in F22, the specific form of $P_{site}$ is not
of importance, but with expected width of the function is. The overview presented here serves as the background to the discussion in the next section

## 5    Continuous or discrete $k(T)$

Considering an observed nucleation event in a drop as a sam-
ple drawn from the distribution $P_{site}$ is fundamental in F22 and it the basis for assuming $k(T)$ to be a continuous function. In Section 3.1 of F22 it is argued that given INPs and sites have site nucleation rates that can yield freezing event over the "entire continuous temperature range". The Gaus-
sian form for $P_{site}$ in V08 aids this argument. This is correct in the abstract but the magnitude of that function is highly centered. Furthermore, as pointed in in Section 4 the form of $P_{site}$ is not well known. The Gaussian in V08 was a convenient way to try to match prediction with observation. Future
work may show a different result for $P_{site}$.

Focusing just on the spread of freezing temperatures resulting from $P_{site}$, in V08 a much narrower spread is postulated while F22 takes the spread to be quite broad. This contrast is a sign of incomplete knowledge. Since an exact value is not needed for the analyses of F22, the focus here
on uncertainty about $P_{site}$ can be viewed as an alert for recognizing what elements are incorporated in the results given in F22. For the great majority of cases, there will be no important consequences. In cases where there are large temperature gaps in the observed freezing temperatures of a set
of drops, neglecting a $P_{site}$ of narrow spread would lead to over-interpretation of the data in that gap using the variable bin widths and assuming continuity.

The use of fixed bin intervals does not exclude that $k(T)$ be derived as a continuous algebraic function by smoothing
and curve fitting step. As to an a priori assumption of continuity and the methods of F22, or the post-hoc fitting of a function are preferred will vary with objectives and styles of analysis.

F22 also makes use of $k(T)$ derived by differentiation of
the cumulative spectrum $K(T)$. If $K(T)$ is a smoothed function, or an algebraic fit than the effect of $P_{site}$ is included and hence the situations is as already discussed.If $K(T)$ is formed by a summation of $k(T)$ over discrete bins, the same considerations apply regarding the appropriateness of fixed
or variables intervals as for $k(T)$ (Section 3). .

## 6    Concluding words

This examination of the differences that arise from wanting the differential spectrum to be data representation, or want-
ing it to also include consideration of random variability of nucleation temperatures on every nucleating site led to looking more closely at what is known and what can be assumed about heterogeneous ice nucleation. In particular, the application of variable bin intervals in data processing was scruti-
nized and shown to have disadvantages for data presentation but useful for the analyses in F22.

This article, and the open discussion associated with it (references here as Fahy and Sullivan, 2023; Petters, 2023 and Vali, 2023), may be helpful to researchers using the dif-
ferential spectra to gain clear understanding of the principles involved.

*Code availability.* The routines used for producing Fig. 1 were written in IDL. The code us available from the author on request.

*Author contributions.* The author did all data processing and writing.                                                                                   95

*Competing interests.* There is no competing interest related to this writing.

*Acknowledgements.* This article benefitted greatly from the review process. Reviewer comments led to broadening the scope of the article and increased the depth of the conclusions. All reviewers are thanked, with special acknowledgements for the contributions by Markus Petters, W.D> Fahy and R.C. Sullivan.

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
