# Peer review of "Comment on "A universally applicable method of calculating confidence bands for ice nucleation spectra derived from droplet freezing experiments" by Fahy, Shalizi and Sullivan (2022)."

_Atmospheric Measurement Techniques, 2023_

## Referee Comment (RC2)

**Review by Markus Petters**

Prof. Vali's comment concerns two claims. First, it is claimed that constant bin intervals are preferred when evaluating differential freezing spectra. Dynamic binning as proposed in Fahy et al. (2022) leads to mathematical artifacts. Second, it is claimed that there is no fundamental reason that the differential spectra are either continuous or monotonically increasing.

The comment brings to the fore a disagreement between Vali (2019) and Fahy et al. (2022) in how to compute the differential freezing spectrum. Most of the subtleties are already discussed in these preceding works. Both Vali (2019, 2023) and Fahy et al. (2022) make valid points. Also, both Vali (2019, 2023) and Fahy et al. (2022) make some claims I don't fully agree with. Ultimately, the differences between the two approaches are minor and, as argued below, the approaches are probably statistically equal once accounting for the confidence limits. Thus the approach adopted may come down to personal preference. Nevertheless, debating these issues is valuable and AMT is an appropriate forum. I therefore support publication.

**Comments**

*First claim. Constant bin intervals are preferred when evaluating differential freezing spectra*

Prof. Vali appears to be on solid statistical ground to reject dynamic binning. It is shown via the illustration in their Figure 1 that the difference is minor/negligible if the data density is high, but potentially substantial when data are sparse. To better understand the underlying arguments, I implemented both algorithms for the dataset given in Table 1 by Vali (2019), which was generated by Polen (2018). Contrary to Vali, the representation is on a linear scale to facilitate the plotting of zero values. However, Vali presents reasonable arguments to show this graph on a logarithmic scale to better visualize low k(T) values.

[Figure]

**Figure 1.** Comparison of k(T) evaluated using the algorithm of Fahy et al. (2022) and Vali (2019, 2023)

Comparing the difference between these implementations, my assessment of the issue is as follows.

(1) The method proposed by Fahy et al. (2022) produces a more noisy k(T). This is acknowledged in Fahy et al. (2022) and proposed to be overcome by applying a smoothing algorithm. The approach was motivated by Fahy et al. (2022) because "binning is widely accepted as an inefficient interpolation method for measurements of continuous variables such as ice nucleation activity, and has been shown to reduce statistical power and bias statistical results in data from a variety of disciplines.". Personally, I am not fully convinced that explicit smoothing is necessarily superior to the implicit smoothing provided by fixed binning. Any significant statistical bias in one or the other method would have to be proven using simulated data with known truth.

(2) The method proposed by Vali et al. (2019, 2023) with constant binning smoothes the spectra. The degree of smoothing depends on the selected bin width ΔT. Vali (2019) provides some discussion about the influence of ΔT on the spectrum and states that "it is recommended to select a suitable value for ΔT and use it for the whole data set.". However, no satisfactory rule is given for how to define suitable. Due to the uncertainty in which ΔT is optimal the method itself remains a moving target when comparing fixed binning to dynamic binning plus smoothing.

(3) The method of Fahy et al. (2022) will never yield zero k(T). As correctly argued by Vali, k(T) = 0 is a valid result that is permitted by constant binning.

(4) The method of Fahy et al. (2022) can yield artificially low results as shown in Figure 1 in Vali (2023). However, it seems to me that this is, in principle, due to the added noise that arises from effectively very narrow binning of the data. The applied smoothing algorithm may alleviate those artificially low values. Furthermore, these artificial low values are essentially due to the interpolation of no data (no freeze events) and one freeze event. The statistical confidence of these points is very low. For a fair comparison, the confidence limits should be given in Figure 1 of Vali (2023) for each data point.

(5) The broad interpretation of the freezing spectra appears to be identical whether the method of Fahy et al. (2022) or Vali (2019,2023) is used.

(6) Lastly, while the statistical details are interesting and important, one should not lose sight of the fact that the differences in the methodology is well within the expected noise of the underlying cold-stage measurements. These include temperature variability across the stage, variability in drop volume, and possible artifacts causing freezing of the drop unrelated to the INP of interest (e.g. substrate artifacts). In my opinion, cold-stage methods are not more precise than ±1 °C. Any binning that is finer than this natural limit may be of limited utility for scientific interpretation. Confidence can only be gained by observing sufficient sample size and number of independent repeats.

In summary, the difference between the two methods is small. I believe once confidence limits are applied, the methods are more or less equal from a statistical perspective. However, I agree with Vali on point (3) and in turn with his conclusion that "there is a reasonable objection to the use of variable ΔT on the basis of principle." Fixed ΔT with a broad enough bin width also

obviates the need to smooth the data. What is not to like? The potential bias in the smoothing from constant binning is likely negligible, but careful statistical work might eventually show otherwise. Finally, given the real-world precision of the underlying measurements, using fixed binning using $\Delta T = 1.0$ °C is the approach we have adopted for cumulative spectra in publications from our own group.

*Second claim. There is no fundamental reason that the differential spectra are either continuous or monotonically increasing.*

Vali argues that based on "Vali (2008) that the site nucleation rate is a steeply rising function over a range of perhaps 1°C, i.e. much smaller than the total range over which freezing events are observed for a set of drops.". This view is strongly supported by subsequent observations e.g. Wright et al. (2013a,b), Herbert et al. (2014), although the temperature range may be a bit wider than the claimed 1°C by Vali (2008). However, whether or not it follows from that that spectra are either continuous or not continuous is not entirely clear. It would seem to me that a given material is composed of a distribution of active sites, each with a site nucleation rate. It would further seem that the combination of a distribution of active sites of different "strength" (characteristic nucleation temperature), and some stochasticity of freezing nucleation for each active site will allow freezing to occur at any value of T. This in turn would argue for a continuous distribution. Even with a continuous distribution, the probability to encounter an active site at temperature T may be vanishingly small so that gaps with zeros in $k(T)$ – especially for discrete sample size – are permitted under this description. I recommend removing the continuous/non-continuous discussion but the point that $k(T) = 0$ is a valid result stands as discussed above.

**References**

Fahy, W. D., C. R. Shalizi, and R. C. Sullivan: A universally applicable method of calculating confidence bands for ice nucleation spectra derived from droplet freezing experiments. Atmos. Meas. Tech., 15, 6819-6836, doi: 10.5194/amt-15-6819-2022, 2022.

Herbert, R. J., Murray, B. J., Whale, T. F., Dobbie, S. J., and Atkinson, J. D.: Representing time-dependent freezing behaviour in immersion mode ice nucleation, Atmos. Chem. Phys., 14, 8501– 8520, doi:10.5194/acp-14-8501-2014, 2014.

Polen, M., Brubaker, T., Somers, J., and Sullivan, R. C.: Cleaning up our water: reducing interferences from non-homogeneous freezing of pure water in droplet freezing assays of ice nucleating particles, Atmos. Meas. Tech., 11, 5315–5334, https://doi.org/10.5194/amt-11-5315-2018, 2018.

Wright, T. P. and Petters, M. D.: The role of time in heterogeneous freezing nucleation, J. Geophys. Res.-Atmos., 118, 3731–3743, https://doi.org/10.1002/jgrd.50365, 2013.

Wright, T. P., Petters, M. D., Hader, J. D., Morton, T., and Holder, A. L.: Minimal cooling rate dependence of ice nuclei activity in the immersion mode, J. Geophys. Res.-Atmos., 118, 1–9, doi:10.1002/jgrd.50810, 2013.

Vali, G.: Repeatability and randomness in heterogeneous freezing nucleation. Atmos. Chem. Phys., 8, 5017-5031. doi:10.5194/acp8-5017-2008, 2008

Vali, G.: Revisiting the differential freezing nucleus spectra derived from drop-freezing experiments: methods of calculation, applications, and confidence limits. Atmos. Meas. Tech., 12, 1219-1231,doi: 10.5194/amt-12-1219-2019, 2019

Vali, G.: Comment on "A universally applicable method of calculating confidence bands for ice nucleation spectra derived from droplet freezing experiments" by Fahy, Shalizi and Sullivan (2022), Atmos. Meas. Tech. Discuss. [preprint], https://doi.org/10.5194/amt-2023-138, in review, 2023.

---

## Referee Comment (RC3)

In Vali (2023; V23), two comments are made on Fahy et al. (2022; F22) regarding calculation and interpretation of ice nucleation spectra.

**Continuous vs. discrete ice nucleation spectra**

We will address the comment on the continuous interpretation of ice nucleation spectra first, as the theory underlying the continuous/discontinuous nature of an ice nucleation spectrum is relevant to the choice of temperature interval.

There is no doubt that everything about practical experimental droplet freezing measurements is discrete, from the individual droplet temperature measurements to calculated freezing spectra from droplet freezing experiments. It is never argued in F22 that this kind of measured spectrum is inherently continuous. Instead, it is argued that the measurements made are sampling from an underlying continuous distribution on the basis of several variables that control ice nucleation in reality. One of the key variables to that continuous nature is the ice active site nucleation rate described in Vali (2008) as a steeply rising continuous function centered around a characteristic temperature. When a droplet freezing experiment is performed, fundamentally we are sampling from the probability distribution function (pdf) derived from that nucleation rate function. This pdf is modelled as a continuous gaussian probability density function as shown in Figure 9 of Vali (2008), although the width of the gaussian is very thin.

In F22 it is argued that because this underlying continuous gaussian pdf describes the probability of freezing of a single ice active site, that ice active site has a *theoretical* probability of freezing at any temperature. That does not mean that in a typical experimental timeframe this ice nucleation site will ever initiate a freezing event – the probability of that occurring may be so arbitrarily low that practically it is zero. However, the underlying probability density function is still continuous, and so there is a differential ice nucleation spectrum $k(T)$ – which represents that underlying probability of freezing as a function of temperature – that is also fundamentally continuous. When many such continuous probability density functions are summed, a differential freezing spectrum such as we observe in experiments is produced. In experiments, we measure discrete variables (frozen fraction versus temperature) to represent this continuous function, and in F22 it is argued that therefore a continuous parametric representation of $k(T)$ is a valid interpretation of the data. It is not argued that differential spectra must be monotonically increasing – continuous functions can be nonmonotonic (although cumulative ice nucleation spectra, $K(T)$, do have to be monotonic). It is also not argued that gaps with 'measured' zeros (i.e. no droplets freeze) are not meaningful – as stated, zeros indicate an arbitrarily low probability of freezing in a temperature region. Such a gap can be clearly observed in mixtures, such as in Figure 5 in Beydoun et al. (2017), where the flat portions of the frozen fraction plots would correspond to a value of $k(T)$ approaching zero. Similarly, in Figure 8 in Vali (2019), the binned points between –10.5 and –12.5 degrees Celsius have confidence intervals overlapping zero and can also be interpreted as a value of $k(T)$ approaching zero.

**Choice of temperature bin width: Fixed or variable?**

To address the comment on the choices of temperature intervals, first we would like to clarify that the variable temperature bin sizes proposed in F22 are only used in the 'splinederiv' and 'binning' methods of calculating ice nucleation spectra. It is not used in the method we recommended, the 'smoothedPCHIP approach', as in that case $k(T)$ is calculated based on the derivative of $K(T)$, the cumulative ice nucleation spectrum. Second, it is correct that the variable temperature bin sizes create values for $k(T)$ that are dependent on neighboring events. While this may not have been the original intention of $k(T)$ as developed

in Vali (1971), in F22 it is argued that this dependence is desirable to better represent the underlying probability distribution that k(T) is meant to measure (and which it is derived from experimentally).

Consider the example data from V23. In the original dataset, k(T) is relatively constant with respect to temperature, except in the beginning of the temperature window considered where it is slightly higher. Based on the model of ice nucleation discussed above, this dataset represents a summation of many probability density functions describing many possible sites with differing characteristic temperatures. We have abstracted this as given as the thick grey line overlaid on Figure 1 from V23 below (thin grey lines show individual gaussians with a width of 1 ºC). When the section in the middle is removed, this removes a subset of these sites, shown as the dashed black line, represented by removing the gaussians between –13 and –15 ºC.[1] While this visual is abstracted, it shows that the large decrease observed from a variable temperature interval may represent the underlying meaning of the removal of points from the spectrum better than the spectrum using fixed temperature bins. Why, then, is it assumed that the points calculated at the edge of the gap using the fixed temperature interval are more correct than those calculated using the variable temperature interval? V23 claims without providing proof or reasoning that because the variable temperature bin point disagrees with the fixed temperature bin point, it is an undesirable artifact. However, we claim it could be the inverse: The variable temperature bin method actually better represents the physical

[Figure]

**Figure 1.** An overlay of Figure 1 from V23 showing how removing a set of droplets (or more specifically, freezing events in a region of the spectrum) would be reflected in a theoretical underlying freezing probability distribution.
* * *
[1]This interpretation of what removing points in the middle of an ice nucleation spectrum means is different from that implied by V23 and relies on the assumption discussed in the first section that ice nucleation spectra have an underlying continuous probability density function derived from the ice nucleation rate function of ice nucleation sites. With this interpretation of ice nucleation, it would be unrealistic to have an infinitely sharp cutoff of ice nucleation activity at -13 and -15 ºC to form this gap, as this would make the nucleation rate function of sites on the edge of this gap differ from those of other sites (by having a sudden cutoff in nucleation rate or sudden jump from zero) and would therefore deviate from the model of ice nucleation being used.

meaning of removing a series of points than the fixed temperature interval method which does not change even though the underlying data has been fundamentally altered.

The reasons behind this difference are threefold. The first is exactly as described in V23: With variable temperature bins, points are not assumed to be independent of their neighbors – rather, the distance between neighboring points is assumed to contain information about the probability of freezing in that region of the temperature spectrum. This makes sense; if a lot of freezing events are observed in a region of a spectrum, it has a high value of k(T), and if few freezing events are observed, it has a low value of k(T). The variable temperature interval reflects this relationship by directly calculating the value of k(T) based on the distance between points and has the additional benefit of maintaining data density (i.e. sampling rate) in regions where many freezing events are observed. This additional data density is useful to increase statistical confidence in these regions of the spectrum. Why, then, are alterations in k(T) due to changes in neighboring freezing effects an undesirable artefact?

We argue that this alteration is actually desirable, for the second reason behind the difference in interpretations: The modified spectrum shown above represents a fundamentally different sample from the original. By artificially removing points in a targeted (i.e., nonrandom) way from a random sample of the probability density function of freezing, the underlying probability density function being sampled from is being changed (i.e.. set to zero between –15 and –13 ºC). With a non-probabilistic measurement, this would not be a problem and the interpretation of the decrease in k(T) in the gap in ice nucleation events presented in V23 would be valid. However, with a probabilistic measurement this becomes a problem. The differential spectrum k(T) relies on equating an empirically observed probability of a droplet freezing at temperature T to an expected value for an unobserved variable – the probability of containing a nucleus that is active at a temperature T (actually the infinitesimal interval from T to T–dT), as shown in Vali (1971) equation 5. Removing the droplets in a given range for the modified spectrum is changing one side of the equation without changing the interpretation of the expected value.

In plain terms, only those droplets that froze between –13 and –15 ºC were removed, ignoring the fact that droplets that froze before –13 ºC could have contained nuclei that would freeze from –13 and –15 ºC, that the droplets that froze between –13 and –15 ºC could have been late (or early) freezers from the nuclei that are active on the edges of either of the remaining ice modes of ice nucleation activity, or that the droplets that remain just before –13 ºC and just after –15 ºC could have been early (or late) freezers from the region of nuclei that was removed. This does not influence the droplets far from the temperature region that was removed, because it was very unlikely they would freeze in the removed region. However, close to the edges of the removed region, the probability that *only* those droplets that froze in the region of interest and *none* of the droplets outside of the region were removed is unlikely. The variable temperature interval partially corrects for this, as the sudden increase in distance between points results in a drop in calculated ice nucleation probability represented by k(T). The fixed temperature interval method as shown does not correct for this.

Third, there is also a nuance hidden in the fixed temperature interval example given in V23: Consider an ice nucleation spectrum similar to the original shown in Figure 1 with a 10x larger population of droplets using the same fixed temperature intervals used above. Again, let the points between –13 and –15 ºC be removed and the spectra recalculated. Instead of 1 or 2 droplets per bin, now there are 10 or 20. Now, instead of the gap in ice nucleation being exactly on the edge of the fixed temperature bins, shift the temperature bins by half their width in either direction. Now, half of the droplets in each of the edge bins have been removed. What is the resulting change to the spectrum? *Exactly the same as that observed using*

*the variable temperature bins.* The bins on either edge of the gap will be reduced by approximately 50%, simply because the frame of reference from which the bin locations have been calculated has shifted. If the reduction in k(T) using the variable temperature bins is an undesirable artifact, surely this would be too? However, we argue it is the inverse: The unchanged edge points when the fixed temperature bins are used is the undesirable artifact, and the variable temperature bin or 'shifted' fixed temperature bins are the more accurate response.

**Final thoughts and caveats**

None of the arguments above should be interpreted to mean that representing an ice nucleation spectrum discretely or using fixed temperature bins for calculating k(T) is invalid. Whether continuous or discrete, fixed or variable temperature intervals, each calculation method makes its own assumptions that must be carefully considered. In F22, it is argued that the continuous interpolations have increased statistical power than discrete binning methods of interpolation, not that the discrete binning methods are inherently incorrect. The reason continuous interpolations are adopted is to facilitate comparisons between ice nucleation spectra and to avoid loss of information about the shape of the ice nucleation spectrum, which coarse binning approaches might obfuscate. This benefit is likely minimal in traditional pipetted droplet-on-substrate approaches, but we posit it will be increasingly useful with microfluidic or similar approaches that analyze large ensembles of droplets. Similarly, it is argued that variable temperature intervals allow for better use of the collected data and avoids loss of information by treating each point separately, not that fixed temperature intervals are incorrect. When experimental uncertainty is considered, the two calculation methods may be statistically insignificantly different as discussed by Markus Petters in his review of V23. If the assumptions made using variable temperature intervals or continuous spectra are incorrect or misaligned with the meaning of ice nucleation spectra, this argument must be clearly presented; it was not in V23.

There is one major caveat that must be considered when variable temperature intervals are used that V23 does not discuss, but Markus Petters does in his review: When there is a lack of freezing events for a long period, it is never explicitly calculated or expressed in that region of the ice nucleation spectrum that the value of k(T) is zero. As such, when variable temperature intervals are used to calculate k(T), the resulting points should be interpreted as measurements of k(T) at a given point of T, not over the entire 'bin' represented by the temperature interval. Indeed, since more than one point is never in a temperature interval at a time, this method is not really a binning approach. Instead, it treats the value $\Delta T$ as a variable, not a bin width. A lack of measurements in a region should then be interpreted literally: there were no freezing events there, so k(T) is essentially zero. A question should then be raised about how wide of a gap truly represents a value of zero, and we propose that this should be related to two factors: (1) the uncertainty in temperature measurements, and (2) the width of the probability density function for a given type of ice nucleation site measured in an experiment. The second variable could be measured using an ice nucleating material such as Snomax® , which is known to have distinctly observable types of ice nucleants (Beydoun et al., 2017).

A minor additional caveat as discussed by Markus Petters in his review of V23 is the increased noise resulting from using variable temperature intervals. This is a side effect of minimizing the loss of information from binning: the noise is essentially information about the random variability within the experiment. As was discussed in F22, noise can be minimized with the use of a smoothing algorithm, which has its own assumptions and can result in the loss of data and in many cases may not be superior to the smoothing provided by the fixed temperature interval binning approach. Noise is not useful, but this does show how variable temperature intervals could be used to avoid information loss in experiments with less

uncertainty and higher density information. If there is nonrandom variability within a small temperature range, a fixed binning approach may miss it while the variable temperature approach will show it clearly. Of course, this usefulness is limited by the uncertainty of a given experimental setup.

We do acknowledge that in the footnote on line 15 of V23, Gabor Vali is correct that there is an error in F22's description of the k(T) equation. It was originally meant to be expressed as $N_0 - N(T)$, not $N(T)$ alone. This error is not present in the actual data or code presented in F22, only in the descriptive text. We will contact the journal to make a correction to F22.

Finally, regardless of whether continuous spectra or variable temperature intervals are used, the methods for determining confidence limits presented in F22 are unchanged. Continuous (but in the case of k(T), not necessarily monotonic) spectra are used for ease of developing statistical tests to compare ice nucleation spectra and to increase statistical power, and variable temperature bins for calculating k(T) are not actually used in the methods recommended, as in that case k(T) is calculated from the derivative of K(T).

William Fahy, University of Toronto (formerly Carnegie Mellon University)

Ryan Sullivan, Carnegie Mellon University

**References**

Beydoun, H., Polen, M., and Sullivan, R. C.: A new multicomponent heterogeneous ice nucleation model and its application to Snomax bacterial particles and a Snomax-illite mineral particle mixture, Atmospheric Chemistry and Physics, 17, 13545–13557, https://doi.org/10.5194/acp-17-13545-2017, 2017.

Fahy, W. D., Shalizi, C. R., and Sullivan, R. C.: A universally applicable method of calculating confidence bands for ice nucleation spectra derived from droplet freezing experiments, Atmos. Meas. Tech., 15, 6819–6836, https://doi.org/10.5194/amt-15-6819-2022, 2022.

Petters, M.: Comment on amt-2023-138, Atmos. Meas. Tech. Discuss. [review] https://doi.org/10.5194/amt-2023-138-RC2, 2023

Vali, G.: Comment on "A universally applicable method of calculating confidence bands for ice nucleation spectra derived from droplet freezing experiments" by Fahy, Shalizi and Sullivan (2022), Atmos. Meas. Tech. Discuss. [preprint], https://doi.org/10.5194/amt-2023-138, in review, 2023.

Vali, G.: Repeatability and randomness in heterogeneous freezing nucleation. Atmos. Chem. Phys., *8*(16), 5017–5031. https://doi.org/10.5194/acp-8-5017-2008, 2008.

Vali, G.: Quantitative Evaluation of Experimental Results on the Heterogeneous Freezing Nucleation of Supercooled Liquids. J. Atmos. Sci., *28*(3), 402–409. https://doi.org/10.1175/1520-0469(1971)028<0402:QEOERA>2.0.CO;2, 1971.

---

## Author Comment (AC3)

**Replies to Reviewer Comment by Fahy and Sullivan (RC3).**

I thank the authors of this review for their detailed explanations and appreciate this opportunity for further discussion about the issues raised in V23. While this exchange is about real differences, there is agreement in that the goal is to further the strength of interpretation of experiments in terms of nucleus spectra for representing the activity of INPs.

The essence of the discussion is the difference in perspectives given in Section 3 of V23, but with the clearer reasoning that emerged from the reviews and responses. The deeper argument behind the apparently simple difference in data processing methods is whether one wants k(T) to concisely represent observed data, or to reveal an "underlying probability of freezing" incorporating additional assumptions about the spread of possible freezing temperatures for each nucleating site ($P_{site}$ in the following). V23 focused on the former, RC3 on the latter perspective. Both are valid goals, and the choice will vary from case to case.

RC3 is copied below, with my responses inserted in offset paragraphs and using a different font.

In Vali (2023; V23), two comments are made on Fahy et al. (2022; F22) regarding calculation and interpretation of ice nucleation spectra.

Continuous vs. discrete ice nucleation spectra

We will address the comment on the continuous interpretation of ice nucleation spectra first, as the theory underlying the continuous/discontinuous nature of an ice nucleation spectrum is relevant to the choice of temperature interval.

Agree that this is relevant but separable from the main point of V23.

There is no doubt that everything about practical experimental droplet freezing measurements is discrete, from the individual droplet temperature measurements to calculated freezing spectra from droplet freezing experiments. It is never argued in F22 that this kind of measured spectrum is inherently continuous. Instead, it is argued that the measurements made are sampling from an underlying continuous distribution on the basis of several variables that control ice nucleation in reality. One of the key variables to that continuous nature is the ice active site nucleation rate described in Vali (2008) as a steeply rising continuous function centered around a characteristic temperature. When a droplet freezing experiment is performed, fundamentally we are sampling from the probability distribution function (pdf) derived from that nucleation rate function. This pdf is modelled as a continuous gaussian probability density function as shown in Figure 9 of Vali (2008), although the width of the gaussian is very thin.

The phrase in F22 which was criticized in V23 is that the spectra are inherently continuous. While the variability due to stochastic variability for given sites is mentioned in the top paragraph of page 6820 in F22, the rest of the paper does not make explicit mention of this (no reference is given to Vali 2008, V08). The variable bin width approach appeared to have been implemented to assure continuity. The reasoning explained in RC3 is welcomed; it helped to clarify the issues.

The site nucleation rate defined in in V08 (Fig. 9 right-hand panel) is a steeply rising function. The expected pdf of freezing temperatures (left-hand panel in that figure) was assumed to be a Gaussian of 0.2 or 0.42°C width for two different samples. There was no a priori reason given in V08 for the Gaussian distribution to be correct. It was used to interpret the re-freezing experiments of V08. It was also argued that the observed pdf results from two contributions, the molecular dynamics of embryo formation (unquestionable) and possible changes in site configurations (unproven). It was also shown that $P_{site}$ may be different for different substrates. These details are relevant to underscore that $P_{site}$ is a yet poorly defined function. Conceptually it is well founded and there is some empirical evidence for its characteristics. Still, it is not out of the question that $P_{site}$ will be shown in the future to be a box function with sharp temperature limits in temperature, and it will almost certainly have rather different magnitudes for mineral and biological INPs. In any case, from what is now known, the range of expected freezing temperatures on a site of given characteristic temperature can be expected to be something of the order of 1°C, although other sources indicated that this range may be larger. It is rather difficult to obtain empirical results for $P_{site}$ and theoretical estimates are unreliable.

For the argument in F22, the precise form of $P_{site}$ is not of importance, but its width is relevant in cases with wide separations of freezing events. The main point here is that knowledge about $P_{site}$ is scant and this uncertainty is carried, even if only to a minor degree, into the methods presented in F22.

In F22 it is argued that because this underlying continuous gaussian pdf describes the probability of freezing of a single ice active site, that ice active site has a *theoretical* probability of freezing at any temperature. That does not mean that in a typical experimental timeframe this ice nucleation site will ever initiate a freezing event – the probability of that occurring may be so arbitrarily low that practically it is zero. However, the underlying probability density function is still continuous, and so there is a differential ice nucleation spectrum k(T) – which represents that underlying probability of freezing as a function of temperature – that is also fundamentally continuous.

This is the crux of the matter. Is the width of $P_{site}$ large enough to justify a general assumption of continuity despite possibly large gaps in k(T)? Does one want k(T) to concisely show the results of an experiment, or present those results combined with an assumption about $P_{site}$? Whether

the advantages that arise assuming continuity (as detailed in F22) are
considered important or a pure representation of data is the goal will be
a decision up to specific cases. For dense data, the difference will be
negligible; the point of the discussion here is to clarify the principles
involved.

When many such continuous probability density functions are summed, a differential freezing spectrum
such as we observe in experiments is produced.  In experiments, we measure discrete variables (frozen
fraction versus temperature) to represent this continuous function, and in F22 it is argued that therefore a
continuous parametric representation of k(T) is a valid interpretation of the data. It is not argued that
differential spectra must be monotonically increasing – continuous functions can be nonmonotonic
(although cumulative ice nucleation spectra, K(T), do have to be monotonic). It is also not argued that gaps
with 'measured' zeros (i.e. no droplets freeze) are not meaningful – as stated, zeros indicate an arbitrarily
low probability of freezing in a temperature region. Such a gap can be clearly observed in mixtures, such
as in Figure 5 in Beydoun et al. (2017), where the flat portions of the frozen fraction plots would correspond
to a value of k(T) approaching zero. Similarly, in Figure 8 in Vali (2019), the binned points between –10.5
and –12.5 degrees Celsius have confidence intervals overlapping zero and can also be interpreted as a value
of k(T) approaching zero.

The presence of zeros in k(T) is clearly a possible outcome in any
experiment. This was part of the reasoning in V23 for avoiding the use of
variable bin width.

**Choice of temperature bin width: Fixed or variable?**

To address the comment on the choices of temperature intervals, first we would like to clarify that
the variable temperature bin sizes proposed in F22 are only used in the 'splinederiv' and 'binning' methods
of calculating ice nucleation spectra. It is not used in the method we recommended, the 'smoothedPCHIP
approach', as in that case k(T) is calculated based on the derivative of K(T), the cumulative ice nucleation
spectrum.

Differentiating K(T) is certainly valid in principle, but in practice
there are issues about what smoothing was used, or what algebraic form is
fitted to obtain (K(T). Strictly speaking, K(T) can be given in stepwise
form, summing k(T) at each interval. Then, the choice of data grouping in
bins dictates dT, and the issue raised in V23 about the choice of bin
widths remains valid. This is only circumvented by smoothing of K(T) or
using an algebraic equation. How this is implemented in the
'smoothedPCHIP' approach is beyond what was addressed in V23. For dense
data, the problem is likely to be imperceptible.

Second, it is correct that the variable temperature bin sizes create values for k(T) that are dependent on
neighboring events.

This was the main point in V23.

While this may not have been the original intention of k(T) as developed in Vali (1971), in F22 it is argued that this dependence is desirable to better represent the underlying probability distribution that k(T) is meant to measure (and which it is derived from experimentally).

A matter of judgement, as said before, whether k(T) is intended to represent the underlying probability distribution, or the observed data which then can always have some smoothing applied or be fitted with algebraic expressions *post hoc*.

Consider the example data from V23. In the original dataset, k(T) is relatively constant with respect to temperature, except in the beginning of the temperature window considered where it is slightly higher. Based on the model of ice nucleation discussed above, this dataset represents a summation of many probability density functions describing many possible sites with differing characteristic temperatures. We have abstracted this as given as the thick grey line overlaid on Figure 1 from V23 below (thin grey lines show individual gaussians with a width of 1 ºC). When the section in the middle is removed, this removes a subset of these sites, shown as the dashed black line, represented by removing the gaussians between –13 and –15 ºC.[1] While this visual is abstracted, it shows that the large decrease observed from a variable temperature interval may represent the underlying meaning of the removal of points from the spectrum better than the spectrum using fixed temperature bins. Why, then, is it assumed that the points calculated at the edge of the gap using the fixed temperature interval are more correct than those calculated using the variable temperature interval? V23 claims without providing proof or reasoning that because the variable temperature bin point disagrees with the fixed temperature bin point, it is an undesirable artifact. However, we claim it could be the inverse: The variable temperature bin method actually better represents the physical
* * *
[1] This interpretation of what removing points in the middle of an ice nucleation spectrum means is different from that implied by V23 and relies on the assumption discussed in the first section that ice nucleation spectra have an underlying continuous probability density function derived from the ice nucleation rate function of ice nucleation sites. With this interpretation of ice nucleation, it would be unrealistic to have an infinitely sharp cutoff of ice nucleation activity at -13 and -15 ºC to form this gap, as this would make the nucleation rate function of sites on the edge of this gap differ from those of other sites (by having a sudden cutoff in nucleation rate or sudden jump from zero) and would therefore deviate from the model of ice nucleation being used.

[Figure]

**Figure 1.** An overlay of Figure 1 from V23 showing how removing a set of droplets (or more specifically, freezing events in a region of the spectrum) would be reflected in a theoretical underlying freezing probability distribution.

meaning of removing a series of points than the fixed temperature interval method which does not change even though the underlying data has been fundamentally altered.

The smooth functions may be judged reliable representations, but in this case, when it is likely that the two regions of the spectrum correspond to two different materials it is less clear how realistic is the dashed curve in the figure.

The reasons behind this difference are threefold. The first is exactly as described in V23: With variable temperature bins, points are not assumed to be independent of their neighbors – rather, the distance between neighboring points is assumed to contain information about the probability of freezing in that region of the temperature spectrum. This makes sense; if a lot of freezing events are observed in a region of a spectrum, it has a high value of k(T), and if few freezing events are observed, it has a low value of k(T). The variable temperature interval reflects this relationship by directly calculating the value of k(T) based on the distance between points and has the additional benefit of maintaining data density (i.e. sampling rate) in regions where many freezing events are observed. This additional data density is useful to increase statistical confidence in these regions of the spectrum. Why, then, are alterations in k(T) due to changes in neighboring freezing effects an undesirable artefact?

Not an artefact *per se*, but representing a processing of the data which may or may not be wanted.

We argue that this alteration is actually desirable, for the second reason behind the difference in interpretations: The modified spectrum shown above represents a fundamentally different sample from the original. By artificially removing points in a targeted (i.e., nonrandom) way from a random sample of the probability density function of freezing, the underlying probability density function being sampled from is being changed (i.e.. set to zero between –15 and –13 ºC). With a non-probabilistic measurement, this would

not be a problem and the interpretation of the decrease in k(T) in the gap in ice nucleation events presented in V23 would be valid. However, with a probabilistic measurement this becomes a problem. The differential spectrum k(T) relies on equating an empirically observed probability of a droplet freezing at temperature T to an expected value for an unobserved variable – the probability of containing a nucleus that is active at a temperature T (actually the infinitesimal interval from T to T–dT), as shown in Vali (1971) equation 5. Removing the droplets in a given range for the modified spectrum is changing one side of the equation without changing the interpretation of the expected value.

The notion of "probabilistic measurement" is debatable, or, at a minimum, an unfortunate expression. Every measurement is a fact (with attached errors). What the measurement means is a matter of interpretation. To me, keeping the two apart is a good thing.

In plain terms, only those droplets that froze between –13 and –15 ºC were removed, ignoring the fact that droplets that froze before –13 ºC could have contained nuclei that would freeze from –13 and –15 ºC, that the droplets that froze between –13 and –15 ºC could have been late (or early) freezers from the nuclei that are active on the edges of either of the remaining ice modes of ice nucleation activity, or that the droplets that remain just before –13 ºC and just after –15 ºC could have been early (or late) freezers from the region of nuclei that was removed. This does not influence the droplets far from the temperature region that was removed, because it was very unlikely they would freeze in the removed region. However, close to the edges of the removed region, the probability that *only* those droplets that froze in the region of interest and *none* of the droplets outside of the region were removed is unlikely. The variable temperature interval partially corrects for this, as the sudden increase in distance between points results in a drop in calculated ice nucleation probability represented by k(T). The fixed temperature interval method as shown does not correct for this.

The width of $P_{site}$ is the relevant measure here. The example described above is made in the abstract, without realistic consideration of the likely shape of the $P_{site}$ function.

Third, there is also a nuance hidden in the fixed temperature interval example given in V23: Consider an ice nucleation spectrum similar to the original shown in Figure 1 with a 10x larger population of droplets using the same fixed temperature intervals used above. Again, let the points between –13 and –15 ºC be removed and the spectra recalculated. Instead of 1 or 2 droplets per bin, now there are 10 or 20. Now, instead of the gap in ice nucleation being exactly on the edge of the fixed temperature bins, shift the temperature bins by half their width in either direction. Now, half of the droplets in each of the edge bins have been removed. What is the resulting change to the spectrum? *Exactly the same as that observed using the variable temperature bins.* The bins on either edge of the gap will be reduced by approximately 50%, simply because the frame of reference from which the bin locations have been calculated has shifted. If the reduction in k(T) using the variable temperature bins is an undesirable artifact, surely this would be too? However, we argue it is the inverse: The unchanged edge points when the fixed temperature bins are used is the undesirable artifact, and the variable temperature bin or 'shifted' fixed temperature bins are the more accurate response.

The choices of the originating value and of the width of temperature intervals for the fixed-interval data presentation are arbitrary and are made with consideration of various experimental factors. Different choices indeed will result in different k(T) data points and this governs the choices of placement and interval width. There are no perfect choices, but

there are adequate ones. Narrowing the gap in the spectrum by shifting the
interval placement, as in the exercise given above, leads to further
points being added in the gap if the variable interval method is used.
That leads back to all the points already discussed.

**Final thoughts and caveats**

 None of the arguments above should be interpreted to mean that representing an ice nucleation spectrum discretely or using fixed temperature bins for calculating k(T) is invalid. Whether continuous or discrete, fixed or variable temperature intervals, each calculation method makes its own assumptions that must be carefully considered. In F22, it is argued that the continuous interpolations have increased statistical power than discrete binning methods of interpolation, not that the discrete binning methods are inherently incorrect. The reason continuous interpolations are adopted is to facilitate comparisons between ice nucleation spectra and to avoid loss of information about the shape of the ice nucleation spectrum, which coarse binning approaches might obfuscate. This benefit is likely minimal in traditional pipetted dropleton-substrate approaches, but we posit it will be increasingly useful with microfluidic or similar approaches that analyze large ensembles of droplets. Similarly, it is argued that variable temperature intervals allow for better use of the collected data and avoids loss of information by treating each point separately, not that fixed temperature intervals are incorrect. When experimental uncertainty is considered, the two calculation methods may be statistically insignificantly different as discussed by Markus Petters in his review of V23. If the assumptions made using variable temperature intervals or continuous spectra are incorrect or misaligned with the meaning of ice nucleation spectra, this argument must be clearly presented; it was not in V23.

Agree with most of the above, except the last sentence. For data
representation, the use of variable bins introduces data errors, as argued
in V23. The question of what meaning is intended for k(T) in a specific
case is another issue.

 There is one major caveat that must be considered when variable temperature intervals are used that V23 does not discuss, but Markus Petters does in his review: When there is a lack of freezing events for a long period, it is never explicitly calculated or expressed in that region of the ice nucleation spectrum that the value of k(T) is zero. As such, when variable temperature intervals are used to calculate k(T), the resulting points should be interpreted as measurements of k(T) at a given point of T, not over the entire 'bin' represented by the temperature interval. Indeed, since more than one point is never in a temperature interval at a time, this method is not really a binning approach. Instead, it treats the value $\Delta T$ as a variable, not a bin width. A lack of measurements in a region should then be interpreted literally: there were no freezing events there, so k(T) is essentially zero. A question should then be raised about how wide of a gap truly represents a value of zero, and we propose that this should be related to two factors: (1) the uncertainty in temperature measurements, and (2) the width of the probability density function for a given type of ice nucleation site measured in an experiment. The second variable could be measured using an ice nucleating material such as Snomax®, which is known to have distinctly observable types of ice nucleants (Beydoun et al., 2017).

 A minor additional caveat as discussed by Markus Petters in his review of V23 is the increased noise resulting from using variable temperature intervals. This is a side effect of minimizing the loss of information from binning: the noise is essentially information about the random variability within the experiment. As was discussed in F22, noise can be minimized with the use of a smoothing algorithm, which has its own assumptions and can result in the loss of data and in many cases may not be superior to the

smoothing provided by the fixed temperature interval binning approach. Noise is not useful, but this does show how variable temperature intervals could be used to avoid information loss in experiments with less uncertainty and higher density information. If there is nonrandom variability within a small temperature range, a fixed binning approach may miss it while the variable temperature approach will show it clearly. Of course, this usefulness is limited by the uncertainty of a given experimental setup.

We do acknowledge that in the footnote on line 15 of V23, Gabor Vali is correct that there is an error in F22's description of the $k(T)$ equation. It was originally meant to be expressed as $N_0 - N(T)$, not $N(T)$ alone. This error is not present in the actual data or code presented in F22, only in the descriptive text. We will contact the journal to make a correction to F22.

Finally, regardless of whether continuous spectra or variable temperature intervals are used, the methods for determining confidence limits presented in F22 are unchanged. Continuous (but in the case of $k(T)$, not necessarily monotonic) spectra are used for ease of developing statistical tests to compare ice nucleation spectra and to increase statistical power, and variable temperature bins for calculating $k(T)$ are not actually used in the methods recommended, as in that case $k(T)$ is calculated from the derivative of $K(T)$.

These discussions in the reviews and responses add to both F22 and V23. As stated in the beginning of this reply, the discussion is not about errors but about two different emphases about what the differential spectra should be. The statistical methods developed in F22 will no doubt help in increasing the value of scientific results that can be represented by the differential nucleus spectra. The purpose in V23, in this reply, and in the final version of V23 is to call attention to a principle difference between the way the spectra are considered in my publications and the way they are in F22, and to alert scientists to the choices they have in constructing and in making use of the spectra.